# Correlation Study Between Dietary Behaviors, Lifestyle, and Psychological Problems in Chinese Children Aged 3–7

**DOI:** 10.3390/nu17010176

**Published:** 2025-01-02

**Authors:** Zixuan Huang, Jiamin Han, Ying Jiang, Shiming Li, Gang Wang, Zhenhe Zhou, Haohao Zhu

**Affiliations:** 1Affiliated Mental Health Center of Jiangnan University, Wuxi 214151, China; 7242808008@stu.jiangnan.edu.cn (Z.H.); 6242828002@stu.jiangnan.edu.cn (J.H.); jiangying1010@jiangnan.edu.cn (Y.J.); mhclsm@163.com (S.L.); zhuhh@jiangnan.edu.cn (H.Z.); 2State Key Laboratory of Food Science and Technology, Jiangnan University, Wuxi 214122, China; wanggang@jiangnan.edu.cn

**Keywords:** dietary behavior, lifestyle, psychological problems, children, influence factors

## Abstract

Objective: This study aimed to assess the psychological health status of children aged 3–7 years in Wuxi and analyze the correlation between dietary behaviors, lifestyle, and psychological problems. Methods: Using a stratified cluster random sampling method, 3–7-year-old children from 18 kindergartens across Wuxi were selected as the survey subjects. An online survey was conducted to collect demographic information about children and their parents, dietary information, lifestyle data, and family backgrounds. Psychological assessments were conducted using the Strengths and Difficulties Questionnaire (SDQ). Results: A total of 3727 preschool children were included. The average SDQ score was 16.12 ± 4.00, with an abnormal rate of 40.81% (1521/3727). After adjusting for various confounding factors, logistic regression analysis indicated that low dietary diversity (OR = 1.299, 95% CI: 1.131–1.492), daily consumption of ultra-processed foods (OR = 1.202, 95% CI: 1.051–1.376), picky eating behavior or ideas occurring more than twice per week (OR = 1.405, 95% CI: 1.227–1.608), engaging in other activities while eating (such as watching TV or playing with toys) more than twice per week (OR = 1.742, 95% CI: 1.510–2.011), lack of a fixed dining position (OR = 1.222, 95% CI: 1.012–1.476), daily screen time exceeding 1 h (OR = 1.353, 95% CI: 1.152–1.590), and daily sleep duration of less than 9 h or more than 13 h (OR = 1.535, 95% CI: 1.262–1.866) were risk factors for psychological problems. Conclusions: The findings highlight the urgent need for targeted interventions addressing insufficient dietary diversity, distracted eating, excessive screen time, and unhealthy sleep habits to improve the psychological well-being of preschool children. Future studies should explore the effectiveness of tailored health and lifestyle interventions for children and their families to mitigate these risks.

## 1. Introduction

Mental health issues in adulthood often manifest as symptoms or psychological problems during childhood [1]. Research indicates that psychopathological mechanisms from childhood, preadolescence, and adolescence persist over time, even though individual symptoms may change [2,3]. It is estimated that children and adolescents represent a significant portion of the global population and are key to future development, with 10–20% affected by mental health problems [4]. Severe mental health issues and behaviors can have profound impacts on the future lives of children, such as school dropout, domestic violence, and self-harm or suicide [5]. Beyond affecting children’s individual growth, mental illnesses or psychiatric disorders impose substantial socioeconomic and healthcare burdens [6]. Attention deficit hyperactivity disorder (ADHD), conduct disorders, and emotional problems are the primary mental health issues among children [7]. A German study reported an emotional and behavioral problem prevalence rate of 10–18% among children aged 4–18 years. Similarly, The Strengths and Difficulties Questionnaire (SDQ) survey conducted in eight Chinese provinces among primary school students revealed a prevalence rate of 11% [8].

Research, both domestic and international, has found that unhealthy lifestyle habits during childhood and adolescence (e.g., lack of physical activity, excessive screen time) are risk factors for children’s mental health problems [9,10,11,12,13]. Studies have demonstrated associations between unhealthy lifestyle behaviors and adverse psychological and behavioral issues in childhood and adolescence [14]. A study in the UK found that children who met the recommended level of physical activity (at least 1 h daily) experienced fewer emotional problems [15]. A systematic review indicated that shorter sleep duration was linked to poorer emotional regulation in children and adolescents [16]. Lifestyle behaviors have become a significant public health concern, as technological advancements and modernized lifestyles have led to an increase in inactivity-related diseases such as obesity and other non-communicable diseases. Besides physical ailments, current research increasingly highlights the interaction between sedentary behavior and mental health outcomes [17,18].

Other studies suggest that dietary patterns and quality are critical factors for mental health in children and adolescents [19,20]. Dietary habits and patterns can significantly reduce psychological problems such as depression and anxiety in children and adolescents, thereby improving mental health [21,22]. Although there is a strong association between poor diet and mood disorders, the evidence supporting these links is inconclusive. However, existing evidence confirms a clear relationship between nutrition and mental health [23]. The composition, structure, and function of the brain depend on the availability of appropriate nutrients, including lipids, amino acids, vitamins, and minerals [24,25]. Thus, food and nutrition influence brain function, making diet an essential factor for mental health, emotional well-being, and cognitive performance [26,27]. A study on dietary habits revealed that high-sugar, high-fat diets prevalent in Western countries are stressors for adolescent mental health [28]. Overconsumption of Western diets is becoming a major health issue [29]. Western diets, characterized by ultra-processed foods rich in saturated fats and sugars, are associated with increased obesity, metabolic disorders, cognitive impairments, and emotional disturbances [30]. Additionally, children and adolescents face mental health challenges during their adaptation to school and social life due to negative influences such as stress, social anxiety, sedentary lifestyles, and unhealthy eating habits [31,32].

It is evident that children’s dietary and lifestyle habits significantly influence the development of mental health issues. Unhealthy diets and sedentary behaviors are modifiable risk factors for mental disorders [33,34]. Evidence suggests that healthy eating and exercise positively affect brain function [35,36]. Therefore, studying the relationship between early dietary patterns, lifestyle habits, and mental health in children is crucial to identify high-risk factors and provide targeted recommendations for interventions. Current research focuses primarily on the mental and emotional problems of school-aged children and adolescents, with little attention given to the mental health status of children aged 3–7 years. Particularly lacking are studies on the long-term impacts of different dietary and lifestyle behaviors on preschool children. Understanding the interaction between diet, lifestyle, and mental health in children and adolescents may guide research on lifestyle-based interventions to improve mental health. Additionally, most studies focus on underdeveloped or rural areas [13,19], with limited research on economically advanced regions in China. This study examines Wuxi City, Jiangsu Province, located in the Yangtze River Basin, with a permanent population of 7.49 million. Wuxi, adjacent to Shanghai, has ranked first in per capita GDP for many years and is representative of economically developed areas in China.

Various tools are used to measure mental health in children and adolescents. The Strengths and Difficulties Questionnaire (SDQ) is a widely used screening tool for children’s mental health problems in over 100 countries across Europe, the Americas, Africa, Asia, and the Middle East [37,38,39,40,41]. The SDQ is widely applied in research and clinical settings due to its quick and easy completion and its strong psychometric properties [42,43]. It is also suitable for assessing emotional and behavioral difficulties in preschool children [43,44]. Furthermore, the parent version of the SDQ is widely accepted by non-health professionals and parents [45].

This study focuses on preclinical research, investigating the impact of dietary behaviors and lifestyle factors on children’s mental health in the developed area of Wuxi City. By conducting detailed surveys of dietary diversity, eating behaviors and habits, physical activity, electronic screen time, and sleep duration among children aged 3–7 years, we examined their correlation with children’s mental health. Based on current evidence, this study confirms the association between dietary and lifestyle behaviors and mental health problems. The findings aim to provide early behavioral interventions for parents and society to prevent severe mental health issues.

## 2. Materials and Methods

### 2.1. Study Subjects

In this study, a survey was conducted among kindergarten children within Wuxi City, Jiangsu Province, using a multistage stratified cluster sampling method. Considering differences in economic development, urban–rural distribution, and survey environments, participants were selected from seven districts (counties): Jiangyin City, Yixing City, Liangxi District, Xishan District, Huishan District, Binhu District, and Xinwu District (Figure 1). After identifying the seven districts, kindergartens within each district were coded using a random number table method. Entire clusters of children from kindergartens were then randomly selected as the survey subjects using the same method.

A total of 18 kindergartens were ultimately included, and 3727 children participated in this study. The research protocol was approved by the Medical Ethics Committee of the Jiangnan University Affiliated Mental Health Center, and informed consent was obtained from all participants.

### 2.2. Survey Method and Content

The questionnaire, designed based on the United Nations Children’s Fund (UNICEF) Multiple Indicator Cluster Surveys (MICS), demonstrated high reliability and validity [46]. Kindergarten teachers were asked to distribute QR codes to parents for the online survey. The survey collected data on demographic information (e.g., child’s gender, age, height, and weight), parental data (e.g., maternal weight, height, education level, age at childbirth, and whether there were any pregnancy complications), and family background (e.g., per capita monthly household income, primary caregiver type). Dietary behavior data included dietary diversity, picky eating, daily consumption of ultra-processed foods, whether the child engaged in other activities during meals (e.g., watching TV, playing with toys), and whether the child had a fixed seat for meals. Lifestyle data included daily outdoor activity time, daily screen time (including TV, smartphones, tablets, etc.), and daily sleep duration. The parent version of the SDQ was used to assess the child’s psychological health [19].

### 2.3. Evaluation Indicators and Definitions

The SDQ comprises 25 items divided into five dimensions, with each item scored on a scale from 0 to 2 (not true, somewhat true, certainly true), corresponding to scores of 0, 1, and 2, respectively. Items 7, 11, 14, 21, and 25 are reverse-scored. The SDQ has a total score range of 0–40, with scores of 0-13 indicating normal psychological status, 14–16 indicating borderline status, and 17–40 indicating abnormal psychological status. In this study, we defined an SDQ score of 17–40 as abnormal and 0-16 as normal. The five dimensions are emotional symptoms, conduct problems, hyperactivity, peer relationship problems, and prosocial behavior, with abnormal scores ranging as follows: emotional symptoms (5–10), conduct problems (4–10), hyperactivity (7–10), peer problems (4–10), and prosocial behavior (4–0). At the same time, we also need to be aware of certain limitations of the SDQ scale. First, the SDQ scale has a degree of subjectivity: When guardians assess children, their subjective perceptions may influence the evaluation, failing to accurately reflect the child’s actual psychological or behavioral problems. Additionally, the scale is developed within a specific cultural context, and different cultural backgrounds may vary in their expectations and definitions of psychological and behavioral norms. Second, the psychological assessment scope is limited: The SDQ scale primarily focuses on five dimensions—emotional symptoms, conduct problems, hyperactivity/inattention, peer relationship problems, and prosocial behavior. It covers little about emerging psychological and behavioral issues, such as internet addiction or gaming addiction. Finally, the SDQ scale cannot be used for a definitive diagnosis: The SDQ scale serves as a screening tool for an initial determination of whether children and adolescents may have psychological or behavioral issues. It cannot provide a conclusive diagnosis of specific psychological disorders like professional clinical evaluations. Furthermore, factors such as external influences may lead to false positives or false negatives.

Dietary diversity was assessed according to the Food and Agriculture Organization (FAO) guidelines for measuring individual dietary diversity. Daily food intake was categorized into nine groups: cereals, tubers, vegetables, fruits, legumes, meat, fish, eggs, and dairy [47]. A dietary diversity score (DDS) was calculated based on the number of food groups consumed in the last 24 h, with a total possible score of 9. We defined DDS ≤4 as low dietary diversity [47]. BMI-z scores were calculated according to the World Health Organization (WHO) standards [48]. BMI-z scores were categorized into four groups: underweight (≤−1), normal weight (−1–1), overweight (1–2), and obese (≥2) [49]. Pregnancy complications included conditions such as hypertension, diabetes, hyperlipidemia, viral hepatitis, hyperthyroidism, and hypothyroidism. Ultra-processed foods included sugary drinks, processed meats, fried foods, sweets, and puffed snacks. Daily consumption of any of these was classified as daily consumption of ultra-processed foods. Picky eating or food avoidance was defined as occurring more than twice a week. Engaging in other activities while eating was also defined as occurring more than twice a week. Based on the Chinese preschoolers’ (ages 3–6) physical activity guidelines [50], outdoor activity time was classified as more or less than 2 h per day. Total sleep time was categorized as 10–13 h/day, and screen time was classified as more or less than 1 h/day.

### 2.4. Quality Control

Before the survey began, the research team members held meetings with kindergarten teachers’ representatives, parents’ representatives, and child healthcare physicians to discuss the feasibility and scientific validity of the survey content. An initial implementation plan was drafted, addressing potential issues during the survey process. A pilot survey was then conducted in a selected kindergarten to identify and address any problems or shortcomings in the survey process. The team reconvened to refine and finalize the implementation plan based on feedback from the pilot survey.

Prior to the survey, all team members underwent rigorous training to ensure consistent standards during the investigation. In the questionnaire design, response options were set for each question to avoid omissions. Considering variations in educational levels and related knowledge among participants, simple and understandable options were used to facilitate comprehension of the survey questions. Logical correction functions were incorporated into the questionnaire to prevent data entry errors.

Each day, designated personnel summarized and reviewed the data, providing feedback during daily quality meetings to analyze and resolve emerging issues promptly. Corrections were implemented in subsequent survey processes. For subjective errors or omissions by respondents, kindergarten teachers were tasked with reminders and assistance. For professional issues that were difficult to explain, the research team members provided one-on-one telephone consultations to clarify and address concerns.

During the data collection and analysis phase, incomplete questionnaires with significant gaps were supplemented to ensure data completeness. Cross-verification of kindergartens and children’s names was conducted to eliminate duplicate or ineligible samples. Data entry was carried out by two individuals independently, and any discrepancies were resolved by a third person.

### 2.5. Statistical Analysis

Statistical analysis was performed using SPSS 19.0 software. Measurement data were expressed as x ± s, and categorical data were expressed as rates. The chi-square test (χ^2^) was used for categorical data analysis. A univariate analysis was conducted for each group of variables, and variables with statistically significant differences were included in a multivariate logistic regression analysis. The presence or absence of psychological problems was used as the dependent variable, while dietary behaviors and lifestyle habits were included as independent variables. The model was used to comprehensively analyze psychological health problems and the influencing factors across different dimensions. *p* < 0.05 was considered statistically significant.

## 3. Results

### 3.1. Basic Information

This survey collected a total of 3727 valid questionnaires from 18 kindergartens, covering children in small, medium, and large classes. The age distribution ranged from 3 to 7 years, with an average age of 5.35 ± 0.98 years. There were 1924 boys (51.62%) and 1803 girls (48.38%). The average SDQ score was 16.12 ± 4.00, with an abnormal psychological rate of 40.81% (1521/3727). In terms of specific dimensions, the emotional symptoms score was 2.39 ± 1.35, with an abnormal rate of 7.62% (284/3727); the conduct problems score was 4.21 ± 1.31, with an abnormal rate of 69.60% (2594/3727); the hyperactivity score was 1.59 ± 1.52, with an abnormal rate of 0.43% (16/3727); the peer relationship problems score was 2.90 ± 1.00, with an abnormal rate of 24.07% (897/3727); and the prosocial behavior score was 5.03 ± 1.43, with an abnormal rate of 34.02% (1268/3727). These results are shown in Table 1.

### 3.2. Analysis of Factors Influencing Psychological Problems in Children

The prevalence of psychological problems among children aged 3–7 years in Wuxi, China, is 40.81% (1521/3727), which is significantly high. This elevated prevalence may be influenced by the limitations of the measurement scale itself. Of course, the impact of current social developments on children’s mental health must also be considered. Therefore, it is essential to focus on longitudinal studies to observe the occurrence of mental health problems in children over time. Univariate analysis revealed that the prevalence of psychological problems in boys is higher than in girls, which is consistent with previous studies. This may be attributed to differences in brain development between genders, variations in parenting styles, and distinct societal expectations during growth, all of which are influenced similarly by parents. Differences in the prevalence of psychological problems were also observed among children with varying BMI-Z scores and mothers with different BMI categories, indicating that nutritional status or obesity levels affect the development of mental health.

Moreover, significant differences in the prevalence of children’s psychological problems were found based on maternal and paternal educational levels, the primary caregiver, and household per capita monthly income. It is widely acknowledged that mental health is directly influenced by educational attainment, economic conditions, and different parenting styles. All the above group differences were statistically significant (*p* < 0.05), as shown in Table 2. In the dimensional analysis, we found the following statistically significant differences: Emotional Problems: differences in children’s BMI-Z scores and fathers’ educational levels (*p* < 0.05). Conduct Problems: differences in children’s ages, mothers’ educational levels, and primary caregiver types (*p* < 0.05). Hyperactivity Symptoms: differences in mothers’ and fathers’ educational levels and family income (*p* < 0.05). Peer Relationship Problems: differences in children’s BMI-Z scores, gender, mothers’ BMI, parental education levels, family income, and primary caregiver types (*p* < 0.05). Prosocial Behavior: differences in children’s gender and family income (*p* < 0.05). See Table 3.

It is evident from these dimensions that parental educational levels and household economic status directly influence children’s mental health issues. It is well known that children’s good mental health status is significantly affected by their parents or guardians. Our research found that children of mothers or fathers with lower educational levels are more likely to exhibit emotional problems. Children from economically disadvantaged households are also more likely to develop psychological problems. Similarly, it is not surprising that differences in nutritional status and gender lead to varying psychological outcomes in children. See Table 3.

### 3.3. Analysis of Factors Influencing Children’s Psychological Problems

#### 3.3.1. Correlation Between Dietary Diversity and Psychological Problems

Compared to children with normal dietary diversity, children with low dietary diversity were more likely to exhibit emotional problems (OR = 1.349, 95% CI: 1.054–1.727), conduct problems (OR = 1.211, 95% CI: 1.044–1.405), peer relationship problems (OR = 1.284, 95% CI: 1.097–1.503), prosocial behavior issues (OR = 1.260, 95% CI: 1.095–1.450), and overall psychological problems (OR = 1.299, 95% CI: 1.131–1.492). These results are presented in Table 4.

#### 3.3.2. Correlation Between Ultra-Processed Food Consumption and Psychological Problems

Compared to children who did not consume ultra-processed foods daily, those who did were more likely to experience psychological problems (OR = 1.202, 95% CI: 1.051–1.376). See Table 4 for more details.

#### 3.3.3. Correlation Between Picky Eating Behavior and Psychological Problems

Children who exhibited picky eating behavior more than twice per week were more likely to experience emotional problems (OR = 1.594, 95% CI: 1.232–2.062), conduct problems (OR = 1.310, 95% CI: 1.137–1.509), hyperactivity (OR = 5.638, 95% CI: 1.241–25.610), peer relationship problems (OR = 1.203, 95% CI: 1.029–1.406), and overall psychological problems (OR = 1.405, 95% CI: 1.227–1.608) compared to children with less frequent picky eating behaviors. See Table 4.

#### 3.3.4. Correlation Between Engaging in Other Activities During Meals and Psychological Problems

Children who engaged in other activities during meals (e.g., watching TV or playing with toys) more than twice per week were more likely to experience emotional problems (OR = 1.831, 95% CI: 1.430–2.346), conduct problems (OR = 1.323, 95% CI: 1.129–1.551), hyperactivity (OR = 5.123, 95% CI: 1.731–15.162), peer relationship problems (OR = 1.524, 95% CI: 1.297–1.790), and overall psychological problems (OR = 1.742, 95% CI: 1.510–2.011) compared to children who did not engage in such activities. See Table 4.

#### 3.3.5. Correlation Between Fixed Dining Positions and Psychological Problems

Compared to children with fixed dining positions, children without fixed positions were more likely to experience psychological problems (OR = 1.222, 95% CI: 1.012–1.476). See Table 4.

#### 3.3.6. Correlation Between Outdoor Activity Time and Psychological Problems

Children who spent less than 2 h per day on outdoor activities were more likely to experience emotional problems (OR = 1.364, 95% CI: 1.025–1.817), conduct problems (OR = 1.320, 95% CI: 1.130–1.541), peer relationship problems (OR = 1.278, 95% CI: 1.074–1.521), and prosocial behavior issues (OR = 1.647, 95% CI: 1.407–1.928) compared to those who spent more time outdoors. See Table 4.

#### 3.3.7. Correlation Between Screen Time and Psychological Problems

Children who spent more than 1 h per day on screens were more likely to experience peer relationship problems (OR = 1.254, 95% CI: 1.046–1.503) and overall psychological problems (OR = 1.353, 95% CI: 1.152–1.590) compared to children who spent less time on screens. See Table 4.

#### 3.3.8. Correlation Between Sleep Duration and Psychological Problems

Children who slept for less than 9 h or more than 13 h per day were more likely to experience emotional problems (OR = 2.048, 95% CI: 1.519–2.763), conduct problems (OR = 1.281, 95% CI: 1.031–1.591), hyperactivity (OR = 3.031, 95% CI: 1.047–8.768), peer relationship problems (OR = 1.421, 95% CI: 1.147–1.761), and overall psychological problems (OR = 1.535, 95% CI: 1.262–1.866) compared to those who slept for 10–13 h per day. See Table 4.

## 4. Discussion

This study conducted a screening of mental health issues among children aged 3–7 years in Wuxi using the SDQ. The results showed an average SDQ score of 16.12 ± 4.00, with an abnormal psychological rate of 40.81% (1521/3727). After adjusting for various confounding factors, we identified several risk factors influencing children’s mental health. These include low dietary diversity, daily consumption of ultra-processed foods, picky or selective eating behaviors occurring more than 2 days per week, engaging in other activities during meals (e.g., watching TV or playing with toys) more than 2 days per week, the absence of a fixed dining seat, screen time exceeding 1 h per day, and daily sleep duration of less than 9 h or more than 13 h. Based on these findings, we recommend that society and parents pay greater attention to these high-risk factors. Practical steps should include enriching dietary diversity, fostering healthy eating habits such as avoiding picky eating and maintaining a fixed dining seat, limiting excessive screen time, and ensuring adequate but not excessive sleep. Targeted interventions should be implemented to prevent serious mental health problems in children.

For the specific dimensions, the abnormal rates were 7.62% for emotional problems, 69.60% for conduct problems, 0.43% for hyperactivity, 24.07% for peer relationship problems, and 34.02% for prosocial behavior. Compared to older school-aged children [45], we found that the average SDQ score (16.12 ± 4.00) for preschool children was lower than that of school-aged children (20.4 ± 6.2). However, the overall abnormal rate was significantly higher than the 4.8% prevalence rate of psychological problems reported in a Danish cohort study of 5–7-year-olds [51]. Other domestic studies reported that the prevalence rates of emotional symptoms, behavioral problems, hyperactivity, peer relationship problems, and poor prosocial behavior were 39%, 27%, 23%, 12%, and 26%, respectively [19]. Except for emotional problems and hyperactivity, our study showed higher rates of behavioral problems, peer relationship issues, and prosocial behavior abnormalities. At this developmental stage, children aged 4–6 years are undergoing rapid physical and psychological growth, during which mental and behavioral issues may gradually manifest. For instance, children may begin to experience difficulties in emotional regulation or peer interactions. Additionally, with the rapid pace of societal development, increasing family economic and social burdens often lead to tense family atmospheres. Frequent parental conflicts, overly harsh discipline, or excessive pampering can contribute to behavioral problems in children. Poor emotional regulation may cause children to vent frustrations or dissatisfaction by exhibiting aggression towards others. Issues such as attention deficits or an inability to sit still are also commonly observed during childhood development. Furthermore, children who lack effective communication skills, a sense of sharing, or a caring and encouraging family environment may find it challenging to build positive relationships with peers, leading to peer relationship problems and deficits in prosocial behavior. Naturally, factors such as the timing of studies, regional differences, and variations in the versions of measurement questionnaires can also impact the results of investigations into children’s mental health, contributing to differences in reported abnormality rates.

Existing research shows that dietary patterns, physical activity, screen time, and sleep duration are significantly correlated with psychological problems in children and adolescents [52,53,54]. However, few studies have examined the impact of lifestyle habits and dietary behaviors on the mental health of preschool children. After adjusting for sociodemographic factors, our study found that low dietary diversity, daily consumption of ultra-processed foods, picky eating behaviors more than twice per week, engaging in other activities during meals, lack of a fixed dining position, excessive screen time (>1 h/day), and insufficient or excessive sleep (<9 or >13 h/day) were risk factors for psychological problems in children.

Children with low dietary diversity may not receive the necessary nutrients for growth and development, which are closely related to cognitive and behavioral outcomes [55], potentially leading to emotional or behavioral issues. Research indicates that ultra-processed foods, particularly those high in fats and sugars, can harm children’s brain development [56]. Excessive consumption of these foods can result in abnormal glucose metabolism, leading to neurological changes and damage to the blood–brain barrier [57].

Picky eating and engaging in other activities during meals are also psychological risk factors, as regular eating behavior is one of the family’s key behavioral norms, which impacts the psychological health of all family members, including children. Research supports this finding [58]. Studies have also shown that picky eaters are more prone to hyperactivity, possibly due to impaired executive function [59]. Children without a fixed dining position may experience irregular eating patterns, which can lead to erratic personalities and, in turn, affect mental health development. Excessive screen time can harm children’s psychological health, as confirmed by a Canadian study that found screen time and sleep duration had a stronger correlation with mental health than physical activity [53]. Our findings align with this, as prolonged screen use in children and adolescents is consistently associated with poorer mental health [60,61]. Excessive screen time can reduce social interaction, impair cognitive function, and increase impulsivity [62], which can exacerbate emotional symptoms like isolation and social anxiety [63].

Our study also found that both insufficient and excessive sleep duration are linked to psychological problems. Adequate sleep is critical for physical health, emotional regulation, and overall well-being [16]. Lack of sleep has been associated with poor social and psychological health outcomes, such as low self-esteem, depression, and poor social support [64]. Sleep duration influences neurotransmitter levels, affecting mood regulation and executive function [65,66]. Sleep is also tied to stress responses, particularly the activation of the hypothalamic–pituitary–adrenal axis and the sympathetic nervous system, which increase the risk of mental health disorders [67]. Although screen time, physical activity, and sleep duration are interrelated, each factor independently influences children’s psychological health and should be observed as such [54].

Our study’s strength lies in its comprehensive analysis of information related to children, mothers, and families, identifying factors associated with children’s mental health beyond the learning pressure environment. By focusing on aspects such as children’s dietary patterns, behavioral habits, and daily routines, we provide a multi-faceted perspective. Moreover, we utilized commonly understood classifications or definitions for dietary diversity, consumption of ultra-processed foods, screen time, and sleep duration, making it convenient for nationwide applications and offering scientific directions for future longitudinal research. This approach also provides valuable scientific insights for caregivers and schoolteachers to design interventions targeting high-risk behaviors related to children’s mental health. However, this study has certain limitations. First, the sample size is relatively small, as it only includes data from one region, overlooking differences in socioeconomic conditions, cultural practices, ethnic backgrounds, and lifestyles. Additionally, all data were reported by caregivers, which introduces potential information and recall biases. Second, as a cross-sectional study, it cannot establish strong causal relationships between children’s mental health issues and associated factors. Lastly, while we investigated children’s screen time, we did not consider the content or context of screen usage (e.g., educational vs. recreational) nor the quality and context of sleep across different ages, which may have influenced the findings.

## 5. Conclusions

This study highlights the significant impact of dietary behaviors and lifestyle habits on the psychological health of Chinese preschool children aged 3–7 years. Key risk factors identified include low dietary diversity, frequent consumption of ultra-processed foods, irregular eating habits (e.g., picky eating, distractions during meals, and lack of fixed dining positions), excessive screen time, and insufficient or excessive sleep. Addressing these modifiable factors through promoting diverse diets, reducing ultra-processed food intake, establishing healthy eating routines, limiting screen time, encouraging physical activity, and ensuring adequate sleep is crucial for supporting children’s mental well-being. These findings provide actionable insights for parents, educators, and policymakers to implement targeted interventions that foster healthy development and prevent psychological problems in young children.

## Figures and Tables

**Figure 1 nutrients-17-00176-f001:**
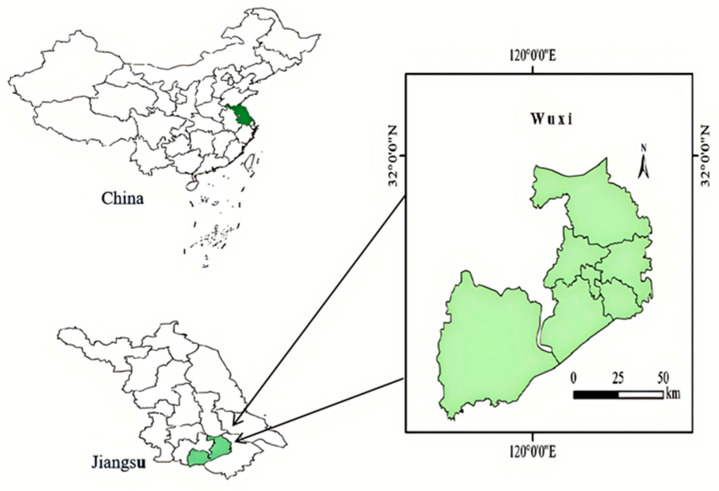
Wuxi map.

**Table 1 nutrients-17-00176-t001:** Psychological health scores and abnormal detection rates in children.

Dimension	Score (x ± s)	Number of Abnormal Cases	Abnormal Rate (%)
Emotional Symptoms	2.39 ± 1.35	284	7.62
Conduct Problems	4.21 ± 1.31	2594	69.60
Hyperactivity	1.59 ± 1.52	16	0.43
Peer Relationship Problems	2.90 ± 1.00	897	24.07
Prosocial Behavior	5.03 ± 1.43	1268	34.02
Total SDQ Score	16.12 ± 4.00	1521	40.81

**Table 2 nutrients-17-00176-t002:** Correlation between psychological health problems (SDQ) and sociodemographic factors in children.

Factors	SDQ				
	Total Number	Abnormal Number	Abnormal Rate (%)	*χ* ^2^	*p*
Sex				10.596	0.001
Male	1924	834	43.347		
Female	1803	687	38.103		
Years				2.350	0.503
3–4	773	329	42.561		
4–5	1304	530	40.644		
5–6	1167	459	39.332		
6–7	483	203	42.029		
BMI-z				18.478	<0.001
≤−1	663	259	39.065		
−1~1	1980	764	38.586		
1~2	408	177	43.382		
≥2	676	321	47.485		
Maternal BMI				16.719	0.001
<18.5	348	126	36.207		
18.5–23.9	2152	840	39.033		
24–27.9	602	265	44.020		
>28	624	290	46.474		
Maternal Age at Birth				0.170	0.895
≤35	3475	1417	40.777		
>35	250	103	41.200		
Pregnancy Complications				1.131	0.288
No	3503	1422	40.594		
Yes	224	99	44.196		
Maternal Education				38.475	<0.001
Elementary and below	19	8	42.105		
Junior high	340	179	52.647		
High school/Vocational	615	273	44.390		
Associate	1067	450	42.174		
Bachelor	1541	560	36.340		
Master or above	145	51	35.172		
Paternal Education				42.244	<0.001
Elementary and below	16	8	50.000		
Junior high	299	163	54.515		
High school/Vocational	658	282	42.857		
Associate	1119	470	42.002		
Bachelor	1475	552	37.424		
Master or above	160	46	28.750		
Household Income (yuan/month)				12.801	0.012
<1000	50	18	36.000		
1000–2000	51	20	39.216		
2000–5000	546	257	47.070		
5000–10,000	1578	648	41.065		
>10,000	1502	578	38.482		
Caregiver type				12.46	0.006
Father	189	79	41.799		
Mother	2311	893	38.641		
Grandparents	1197	536	44.779		
Nanny/Other	30	13	43.333		

**Table 3 nutrients-17-00176-t003:** Correlation between psychological problem dimensions and sociodemographic factors in children.

Factors	Total Number	Emotional Problems				Conduct Problems				Hyperactivity				Peer Relationship Problems				Prosocial Behavior			
		Abnormal Number	Abnormal Rate (%)	*χ* ^2^	*p*	Abnormal Number	Abnormal Rate (%)	*χ* ^2^	*p*	Abnormal Number	Abnormal Rate (%)	*χ* ^2^	*p*	Abnormal Number	Abnormal Rate (%)	*χ* ^2^	*p*	Abnormal Number	Abnormal Rate (%)	*χ* ^2^	*p*
Sex				0.175	0.675			0.42	0.837			3.516	0.061			6.772	0.009			6.004	0.014
Male	1924	150	7.80			1342	69.75			12	0.62			497	25.83			690	35.86		
Female	1803	134	7.43			1252	69.44			4	0.22			400	22.19			578	32.06		
Age				1.078	0.782			41.806	0.001			3.702	0.295			1.277	0.735			12.527	0.006
3–4	773	55	7.12			584	75.55			4	0.52			182	23.54			293	37.90		
4–5	1304	95	7.29			940	72.09			3	0.23			304	23.31			450	34.51		
5–6	1167	95	8.14			780	66.84			8	0.69			288	24.68			388	33.25		
6–7	483	39	8.07			290	60.04			1	0.21			123	25.47			137	28.36		
BMI-z				12.615	0.006			0.864	0.834			0.77	0.994			22.422	0.001			1.337	0.720
≤−1	663	54	8.14			466	70.29			3	0.45			167	25.19			238	35.90		
−1~1	1980	126	6.36			1367	69.04			8	0.40			420	21.21			669	33.79		
1~2	408	45	11.03			290	71.08			2	0.49			110	26.96			136	33.33		
≥2	676	59	8.73			471	69.67			3	0.44			200	29.59			225	33.28		
Maternal BMI				5.639	0.131			1.063	0.786			2.35	0.503			8.432	0.038			5.75	0.124
<18.5	348	27	7.76			239	68.68			0	0.00			75	21.55			117	33.62		
18.5–23.9	2152	149	6.92			1490	69.24			10	0.46			496	23.05			755	35.08		
24–27.9	602	47	7.81			429	71.26			2	0.33			149	24.75			208	34.55		
>28	624	61	9.78			435	69.71			4	0.64			176	28.21			187	29.97		
Maternal Age at Birth				1.559	0.212			0.72	0.396			0.86	0.54			0.082	0.775			0.927	0.336
≤35	3475	270	7.77			2424	69.76			14	0.40			834	24.00			1175	33.81		
>35	250	14	5.60			168	67.20			2	0.80			62	24.80			92	36.80		
Pregnancy Complications				0.580	0.446			0.42	0.559			0.002	0.968			0.031	0.861			0.816	0.366
No	3503	264	7.54			2442	69.71			15	0.43			842	24.04			1198	34.20		
Yes	224	20	8.93			152	67.86			1	0.45			55	24.55			70	31.25		
Maternal Education				10.738	0.057			11.971	0.035			16.068	0.007			41.176	0.001			4.878	0.431
Elementary and below	19	3	15.79			12	63.16			1	5.26			9	47.37			10	52.63		
Junior high	340	36	10.59			240	70.59			3	0.88			116	34.12			122	35.88		
Highschool/Vocational	615	55	8.94			425	69.11			2	0.33			168	27.32			214	34.80		
Associate	1067	80	7.50			762	71.42			7	0.66			259	24.27			365	34.21		
Bachelor	1541	99	6.42			1040	67.49			3	0.19			310	20.12			513	33.29		
Master or above	145	11	7.59			115	79.31			0	0.00			35	24.14			44	30.34		
Paternal Education				15.534	0.008			5.702	0.336			19.803	0.001			40.252	0.001			3.071	0.689
Elementary and below	16	3	18.75			13	81.25			1	6.25			7	43.75			7	43.75		
Junior high	299	29	9.70			210	70.23			3	1.00			110	36.79			96	32.11		
High school/Vocational	658	49	7.45			435	66.11			0	0.00			172	26.14			235	35.71		
Associate	1119	104	9.29			787	70.33			7	0.63			257	22.97			366	32.71		
Bachelor	1475	92	6.24			1039	70.44			5	0.34			324	21.97			510	34.58		
Master or above	160	7	4.38			110	68.75			0	0.00			27	16.88			54	33.75		
Household Income (yuan/month)				4.557	0.336			7.174	0.127			27.467	0.001			24.154	0.001			11.827	0.019
<1000	50	4	8.00			38	76.00			2	4.00			19	38.00			22	44.00		
1000–2000	51	1	1.96			33	64.71			0	0.00			17	33.33			20	39.22		
2000–5000	546	46	8.42			402	73.63			7	1.28			158	28.94			182	33.33		
5000–10,000	1578	129	8.17			1097	69.52			3	0.19			391	24.78			574	36.38		
>10,000	1502	104	6.92			1024	68.18			4	0.27			312	20.77			470	31.29		
Caregiver type				5.627	0.131			6.485	0.09			6.398	0.094			15.801	0.001			2.188	0.534
Father	189	15	7.94			138	73.02			3	1.59			55	29.10			70	37.04		
Mother	2311	158	6.84			1575	68.15			9	0.39			515	22.28			797	34.49		
Grandparents	1197	108	9.02			858	71.68			4	0.33			314	26.23			390	32.58		
Nanny/Other	30	3	10.00			23	76.67			0	0.00			13	43.33			11	36.67		

**Table 4 nutrients-17-00176-t004:** Logistic regression analysis of dietary behaviors, lifestyle habits, and psychological problems in children.

Factors	Emotional Problems ^a^		Conduct Problems ^b^		Hyperactivity ^c^		Peer Relationship Problems ^d^		Prosocial Behavior ^e^		SDQ ^f^	
	OR (95% CI)	*p*	OR (95% CI)	*p*	OR (95% CI)	*p*	OR (95% CI)	*p*	OR (95% CI)	*p*	OR (95% CI)	*p*
Low dietary diversity	1.349 (1.054–1.727)	0.017	1.211 (1.044–1.405)	0.012	2.327 (0.825–6.566)	0.110	1.284 (1.097–1.503)	0.002	1.260 (1.095–1.450)	0.001	1.299 (1.131–1.492)	0.001
Daily consumption of ultra-processed foods	1.049 (0.820–1.342)	0.704	1.081 (0.937–1.246)	0.285	1.090 (0.394–3.016)	0.868	1.083 (0.927–1.264)	0.314	0.921 (0.803–1.057)	0.241	1.202 (1.051–1.376)	0.007
Picky eating behavior	1.594 (1.232–2.062)	0.001	1.310 (1.137–1.509)	0.001	5.638 (1.241–25.610)	0.025	1.203 (1.029–1.406)	0.020	1.117 (0.973–1.281)	0.115	1.405 (1.227–1.608)	0.001
Engaging in other activities during meals	1.831 (1.430–2.346)	0.001	1.323 (1.129–1.551)	0.001	5.123 (1.731–15.162)	0.003	1.524 (1.297–1.790)	0.001	1.034 (0.893–1.198)	0.655	1.742 (1.510–2.011)	0.001
No fixed dining position	1.201 (0.868–1.660)	0.268	1.151 (0.937–1.414)	0.180	0.416 (0.076–2.294)	0.314	1.356 (1.103–1.666)	0.004	1.138 (0.943–1.372)	0.177	1.222 (1.012–1.476)	0.037
Outdoor activity time <2 h/day	1.364 (1.025–1.817)	0.033	1.320 (1.130–1.541)	0.001	0.717 (0.254–2.018)	0.528	1.278 (1.074–1.521)	0.006	1.647 (1.407–1.928)	0.001	1.067 (0.921–1.236)	0.389
Screen time >1 h/d	1.138 (0.854–1.516)	0.377	1.110 (0.932–1.322)	0.242	1.595 (0.531–4.787)	0.405	1.254 (1.046–1.503)	0.014	0.966 (0.818–1.141)	0.682	1.353 (1.152–1.590)	0.001
Sleep duration <9 h or >13 h	2.048 (1.519–2.763)	0.001	1.281 (1.031–1.591)	0.026	3.031 (1.047–8.768)	0.041	1.421 (1.147–1.761)	0.001	1.096 (0.897–1.339)	0.370	1.535 (1.262–1.866)	0.001

Notes: ^a^: Adjusted for BMI-Z and father’s education level. ^b^: Adjusted for age, mother’s education, and primary caregiver type. ^c^: Adjusted for parental education and household income. ^d^: Adjusted for child’s BMI-Z, gender, mother’s BMI, parental education, and household income. ^e^: Adjusted for child’s gender and household income. ^f^: Adjusted for child’s BMI-Z, gender, current BMI, parental education, and household income.

## Data Availability

The dataset generated and analyzed during the current study is available from the corresponding author on reasonable request.

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
