# Peer review of "Correlation Study Between Dietary Behaviors, Lifestyle, and Psychological Problems in Chinese Children Aged 3–7"

_nutrients, 2025, doi:10.3390/nu17010176_

Round 1
Reviewer 1 Report
Comments and Suggestions for Authors
Dear authors, thank you very much for allowing me to read your research. The topic is really interesting. I propose some points of improvement that will give more depth to the study.
Abstract
Reconstruction based on all suggestions.
1. Introduction
The authors establish a convincing common thread. Indeed, it is essential to be able to detect mental health problems in childhood in order to prevent their development in adulthood.
Suggestions: What knowledge gap does your research aim to fill, are there similar studies in China, and what are the differential aspects of your research with respect to previous studies?
It is important to contextualize the problem that this research intends to address in China and specifically in the city of Wuxi. I suggest constructing a section 1.1 entitled contextualization of the research.
2. Materials and Methods
2.1. Study Subjects
Suggestions: It is important that the authors explain in depth the type of sampling used. why does a stratified cluster random sampling? what does this type of sampling consist of?
2.2 Survey Method and Content
Suggestions: Did the parents sign any kind of informed consent? how was the confidentiality of the data guaranteed? did the study go through an ethics committee? how did the authors address common method bias?
2.3 Evaluation Indicators and Definitions
Suggestions: It is important to specify the possible weaknesses or limitations of the scale used (SDQ). It is important to specify the possible limitations of the questionnaire used.
2.4 Quality Control
No suggestions
2.5 Statistical Analysis
Suggestions: Specify in more depth the statistical analyses used. Justify their use.
3. Results
3.1 Basic Information
No suggestions
3.2. Analysis of Factors Influencing Psychological Problems in Children
No suggestions
3.3. Analysis of Factors Influencing Children's Psychological Problems
3.3.1. Correlation Between Dietary Diversity and Psychological Problems
No suggestions
3.3.2. Correlation Between Ultra-Processed Food Consumption and Psychological Problems
3.3.3. Correlation Between Picky Eating Behavior and Psychological Problems
3.3.4. Correlation Between Engaging in Other Activities During Meals and Psychological Problems
3.3.5. Correlation Between Fixed Dining Positions and Psychological Problems
3.3.6. Correlation Between Outdoor Activity Time and Psychological Problems
3.3.7. Correlation Between Screen Time and Psychological Problems
3.3.8. Correlation Between Sleep Duration and Psychological Problems
All correlations are well specified.
4. Discussion
Suggestions: The discussion usually begins with the objective of the research, in this case, to provide early behavioral intervention recommendations for parents and society to prevent the development of serious psychological health problems in children.
The study showed higher rates of conduct problems, peer relationship problems and prosocial behavior abnormalities. It is important to further explore these important differences.
It is important to delve more deeply into the strengths of the study. It is important to include future research. It is a priority for the authors to construct an independent practical implications section oriented to the objectives of the journal and taking into account some articles published in Nutrients. The practical implications of a research are possibly the most important section.
5. Conclusions
Suggestions: Improve the conclusions based on all suggestions.
Author Response
Comments 1: The authors establish a convincing common thread. Indeed, it is essential to be able to detect mental health problems in childhood in order to prevent their development in adulthood. Suggestions: What knowledge gap does your research aim to fill, are there similar studies in China, and what are the differential aspects of your research with respect to previous studies? It is important to contextualize the problem that this research intends to address in China and specifically in the city of Wuxi. I suggest constructing a section 1.1 entitled contextualization of the research.
Response 1: Thank you for your valuable suggestions. We have enriched the introduction by conducting further searches and citing 19 additional relevant studies to thoroughly explain the current state of research both domestically and internationally. We found that similar international studies often focus on single factors affecting mental or behavioral problems in minors, such as dietary habits, sedentary behavior, lack of exercise, screen time, or sleep duration. However, these factors are interdependent and should be considered collectively. Additionally, research on young children's mental health is relatively scarce, with most studies focusing on school-aged children or adolescents due to their higher survey compliance. Our research targets 3-7-year-old children and comprehensively examines the impact of dietary behaviors and lifestyle habits on their mental health, providing scientific recommendations for high-risk interventions in relatively developed areas in China. Furthermore, we introduced the demographic, economic, and geographical characteristics of Wuxi in the introduction, emphasizing the importance of addressing local children's mental health issues. Detailed revisions are available in the updated manuscript under Section 1.1.
Comments 2: It is important that the authors explain in depth the type of sampling used. why does a stratified cluster random sampling? what does this type of sampling consist of?
Response 2: Thank you for your insightful comment. We have provided a detailed explanation of the sampling method in the manuscript. Specifically, we employed a multi-stage stratified cluster random sampling method. Based on economic development, urban-rural distribution, and survey environments, we selected seven districts in Wuxi: Jiangyin, Yixing, Liangxi, Xishan, Huishan, Binhu, and Xinwu. In each district, kindergartens were coded using a random number table, and entire clusters of children from selected kindergartens were included in the study. Ultimately, 18 kindergartens and 3,727 children were surveyed. This method minimizes selection bias by ensuring a representative sample across various socioeconomic and geographic contexts.
Comments 3: Did the parents sign any kind of informed consent? How was the confidentiality of the data guaranteed? Did the study go through an ethics committee? How did the authors address common method bias?
Response 3: Yes, the study received approval from the Medical Ethics Committee of the Affiliated Mental Health Center of Jiangnan University. All participants provided informed consent. Confidentiality was strictly maintained following our institution's academic guidelines, and data were used solely for research purposes. To address common method bias, the research team held pre-survey meetings with kindergarten teachers, parent representatives, and pediatricians to refine the survey design. A pilot survey was conducted to identify potential issues, followed by adjustments to ensure clarity and comprehensiveness. Data quality was ensured through rigorous training, logical checks in the questionnaire, daily data reviews, and error correction mechanisms. The measures are elaborated in the revised manuscript.
Comments 4: It is important to specify the possible weaknesses or limitations of the scale used (SDQ). It is important to specify the possible limitations of the questionnaire used.
Response 4: Thank you for highlighting this point. We have included a detailed discussion on the limitations of the SDQ scale:
Subjectivity: Parental assessments may not accurately reflect children’s true psychological or behavioral problems due to subjective bias.
Cultural Background: The scale, developed in specific cultural contexts, may not fully align with different cultural expectations and definitions of mental and behavioral health.
Limited Scope: The SDQ focuses on dimensions like emotional symptoms, conduct problems, hyperactivity/inattention, peer relationship problems, and prosocial behavior but lacks coverage of emerging issues such as internet or gaming addiction.
Screening Tool: The SDQ serves as a preliminary screening instrument and cannot provide definitive clinical diagnoses. It may produce false positives or negatives due to influencing factors.
These limitations are now explicitly mentioned in the updated manuscript.
Comments 5: Specify in more depth the statistical analyses used. Justify their use.
Response 5: We have added a detailed explanation of the statistical methods used. Specifically, we applied multivariate logistic regression analysis to explore the effects of multiple factors on children's mental health. This method is appropriate for binary dependent variables and helps mitigate confounding factors. By including statistically significant variables from univariate analyses, multivariate logistic regression provides a clearer understanding of independent associations between variables and mental health outcomes. The updated manuscript now includes these justifications and a clearer explanation of the analytical process.
Comments 6: The discussion usually begins with the objective of the research, in this case, to provide early behavioral intervention recommendations for parents and society to prevent the development of serious psychological health problems in children. The study showed higher rates of conduct problems, peer relationship problems and prosocial behavior abnormalities. It is important to further explore these important differences. It is important to delve more deeply into the strengths of the study. It is important to include future research. It is a priority for the authors to construct an independent practical implications section oriented to the objectives of the journal and taking into account some articles published in Nutrients. The practical implications of a research are possibly the most important section.
Response 6: In the revised manuscript, we have restructured the discussion to begin with the research objectives, emphasizing early behavioral interventions for parents and society. Significant findings, such as higher rates of conduct problems and prosocial behavior abnormalities, have been explored in greater depth. We have highlighted the study’s strengths, including a large sample size and comprehensive consideration of dietary and lifestyle factors. Practical implications, such as recommendations for parents and policymakers, have been outlined in an independent section. Future research directions, including longitudinal and multicenter studies, are also discussed.
Comments 7: Improve the conclusions based on all suggestions.
Response 7: The conclusions have been revised to reflect the improvements made throughout the manuscript. They emphasize the study's contributions to understanding the interplay of dietary and lifestyle factors on young children's mental health and outline future directions for research and intervention strategies.
Reviewer 2 Report
Comments and Suggestions for Authors
This study provides interesting insights into the psychological health of preschool children and highlights significant risk factors related to dietary behaviors and lifestyle. With 3,727 children surveyed, the study offers robust data for analysis.
Although the study used stratified cluster random sampling, the sample was limited to children in Wuxi. This limits the generalizability of the findings to other regions with differing socioeconomic, cultural, and dietary norms.
The criteria for selecting the 18 kindergartens are not detailed, which may have introduced selection bias.
The study's cross-sectional nature prevents causal inferences. While correlations between dietary and lifestyle factors and psychological health are identified, it cannot determine whether these factors caused the psychological issues or were consequences of them.
While the SDQ is a well-validated tool, relying solely on it may not capture the full spectrum of psychological health issues. Supplementary qualitative data or clinical evaluations could provide a more nuanced understanding.
The reported abnormal rate of 40.81% is notably high. This raises questions about the appropriateness of the SDQ scoring thresholds in this population or the potential overestimation of psychological issues.
The manuscript does not explicitly define or quantify dietary diversity, which can vary across cultures and individual perceptions.
Screen time exceeding 1 hour daily is flagged as a risk factor, but the content and context of screen use (e.g., educational vs. recreational) are not considered, which could influence the psychological outcomes.
While unhealthy sleep patterns are identified as a risk, the broad range (less than 9 or more than 13 hours) lacks context regarding age-specific sleep needs and quality.
While the study adjusts for "various confounding factors," the specific variables included in the model are not clearly outlined. Omitting critical confounders (e.g., parental mental health, family income) could bias results.
Some of the ORs (e.g., 1.202 or 1.222) indicate weak associations. While statistically significant, these findings may have limited practical relevance.
The reliance on online self-reported surveys introduces the risk of recall bias or socially desirable responses, particularly regarding sensitive information like screen time and dietary habits.
Tracking changes over time would provide better insights into the dynamic relationships between lifestyle factors and psychological health.
Author Response
Comments 1: Although the study used stratified cluster random sampling, the sample was limited to children in Wuxi. This limits the generalizability of the findings to other regions with differing socioeconomic, cultural, and dietary norms.
Response 1: We acknowledge these limitations and have explicitly mentioned them in the manuscript. The discussion now includes plans for future longitudinal and multicenter studies to enhance generalizability and address causal relationships.
Comments 2: The criteria for selecting the 18 kindergartens are not detailed, which may have introduced selection bias.
Response 2: We have clarified the selection criteria. Kindergartens were randomly selected from the seven districts using a stratified cluster sampling method, ensuring representation across socioeconomic and geographic contexts. Details are included in the revised manuscript.
Comments 3: The study's cross-sectional nature prevents causal inferences. While correlations between dietary and lifestyle factors and psychological health are identified, it cannot determine whether these factors caused the psychological issues or were consequences of them.
Response 3: We recognize that the cross-sectional nature of the study only identifies correlations between dietary and lifestyle factors and psychological health. It cannot determine causation, such as whether these factors caused the psychological issues or were consequences of them. This limitation has been explicitly noted in the revised discussion.
Comments 4: The reported abnormal rate of 40.81% is notably high. This raises questions about the appropriateness of the SDQ scoring thresholds in this population or the potential overestimation of psychological issues.
Response 4: As noted in Reviewer 1's response, we have included a detailed discussion of the SDQ’s limitations in the revised manuscript.
Comments 5: The manuscript does not explicitly define or quantify dietary diversity, which can vary across cultures and individual perceptions. Screen time exceeding 1 hour daily is flagged as a risk factor, but the content and context of screen use (e.g., educational vs. recreational) are not considered, which could influence the psychological outcomes. While unhealthy sleep patterns are identified as a risk, the broad range (less than 9 or more than 13 hours) lacks context regarding age-specific sleep needs and quality.
Response 5: Dietary diversity was defined based on FAO guidelines, categorizing daily food intake into nine groups. Screen time and sleep pattern considerations have been acknowledged as limitations and potential directions for future research. Details are provided in the updated manuscript.
Comments 6: While the study adjusts for "various confounding factors," the specific variables included in the model are not clearly outlined. Omitting critical confounders (e.g., parental mental health, family income) could bias results. Some of the ORs (e.g., 1.202 or 1.222) indicate weak associations. While statistically significant, these findings may have limited practical relevance.
Response 6: We conducted univariate analyses on various factors, and variables with statistical significance were included in the model for further analysis. However, some key factors, such as parental mental health, were not included in the scope of this survey. We plan to measure these factors in future studies based on the reviewers' suggestions. Additionally, some OR values indicate weak associations. Although they are statistically significant, the practical relevance of these findings may be limited, and we have addressed this in the discussion.
Comments 7: The reliance on online self-reported surveys introduces the risk of recall bias or socially desirable responses, particularly regarding sensitive information like screen time and dietary habits.
Response 7: To address issues of recall bias or deliberate concealment of sensitive information, we implemented rigorous quality control measures to minimize such occurrences. Before the survey began, the research team held meetings with representatives of kindergarten teachers, parents, and pediatricians to discuss the feasibility and scientific validity of the survey content. These discussions helped formulate the initial survey plan by identifying and addressing potential issues during the survey process. A pilot study was then conducted in one kindergarten to summarize and address any problems or deficiencies. The team held follow-up meetings to finalize a comprehensive survey implementation plan. Prior to the formal survey, team members underwent strict training to ensure standardized procedures during the data collection process. In the questionnaire design, we set clear and simple response options to minimize missing responses and account for varying levels of education and knowledge among participants, ensuring better understanding of the survey questions. Logical validation features were also embedded in the questionnaire to prevent data entry errors. We designated personnel to compile and review data daily, holding quality control meetings to analyze issues and implement corrections promptly in subsequent data collection phases. For subjective errors or incomplete responses, kindergarten teachers were instructed to remind and assist participants. For professional or technical questions, team members provided one-on-one telephone explanations and guidance. During the data collection and analysis phases, we addressed incomplete questionnaires by supplementing missing information to ensure data completeness. Each kindergarten and child’s name were cross-checked to eliminate duplicate or out-of-scope samples. For data entry, two team members independently entered and organized the data, with discrepancies resolved by a third party. These measures ensured the reliability and accuracy of the collected data.
Comments 8: Tracking changes over time would provide better insights into the dynamic relationships between lifestyle factors and psychological health.
Response 8: Thank you for this insightful suggestion. We acknowledge that a longitudinal approach would provide a more comprehensive understanding of the dynamic relationships between lifestyle factors and psychological health. While our current study is cross-sectional in design, we have noted this limitation in the manuscript and outlined plans for future longitudinal studies to track these changes over time and better establish causal relationships. This has been incorporated into the discussion and future research directions sections.
Reviewer 3 Report
Comments and Suggestions for Authors
Dear authors, I believe your research can be relevant enough to be considered for publication in Nutrients. However, I encourage you to make the following revisions:
You need to mention the study’s objectives and future perspectives in the abstract.
The Introduction is too brief. You need to provide a more robust background in this section with an international perspective to better justify the need for your research. You should cite more studies similar to yours and point out the relevance and novelty of your study.
Can you please provide a map area where the study was developed?
Was the applied questionnaire adapted to the study population? Can you please clarify it?
Provide more details about the quality control (section 2.4).
Tables need to be better explained in the text, especially Tables 2 and 3.
Discussion and Conclusions are fine.
Author Response
Comments 1: You need to mention the study’s objectives and future perspectives in the abstract.
Response 1: The abstract has been revised to explicitly state the study’s objectives and outline future research directions.
Comments 2: The Introduction is too brief. You need to provide a more robust background in this section with an international perspective to better justify the need for your research. You should cite more studies similar to yours and point out the relevance and novelty of your study.
Response 2: The introduction has been expanded with additional references and a broader international perspective to better justify the research’s relevance and novelty.
Comments 3: Can you please provide a map area where the study was developed?
Response 3: A map of Wuxi, illustrating the study area, has been included in the revised manuscript.
Comments 4: Was the applied questionnaire adapted to the study population? Can you please clarify it?
Response 4: The SDQ questionnaire was validated for use in China and is widely accepted in diverse cultural settings. This information, along with relevant references, has been added to the revised manuscript.
Comments 5: Provide more details about quality control (section 2.4).
Response 5: Details on quality control measures, including training, pilot testing, logical checks, and error correction mechanisms, have been added to Section 2.4 of the revised manuscript.
Comments 6: Tables need to be better explained in the text, especially Tables 2 and 3.
Response 6: The text has been revised to include a detailed explanation of Tables 2 and 3, highlighting significant findings and their implications.
Round 2
Reviewer 1 Report
Comments and Suggestions for Authors
Dear Authors,
Thank you very much for allowing me to review this second version of the article. I have carefully read this new version, and it has important improvements. First, the authors cite nineteen additional studies to explain the knowledge gap that their research aims to fill. Including new information explains in detail the relevance of this article to previous studies. The new findings enrich the research objective. In fact, the focus of this article on children aged three to seven years is one of its main strengths.
Second, the authors improve the materials and methods section. Specifically, the authors explain in detail the multistage stratified cluster random sampling method and its potential benefits. Third, the authors confirm that the study went through an ethics committee. In addition, the authors explain in detail how they addressed the common bias method through a pilot survey.
Fourth, the authors include a detailed discussion of the limitations of the SDQ scale. Fifth, the authors further specify the statistical analyses used. They incorporate an explanation that adequately justifies their use. Sixth, the authors restructure the discussion of results and construct a separate section on practical implications. Finally, the authors revise and improve the conclusions.
A very good job. Congratulations.
Author Response
Thank you for your considerations.
Reviewer 2 Report
Comments and Suggestions for Authors
After these major modifications in the manuscript, and with the clarifications made by the authors, I believe that the paper can be published.
Author Response
Thank you for your considerations.